# SAMT: Generating Structured Avatar Meshes and Textures from a Single Image

**Muyu Wang** [* 1]  **Jianzhe Gao** [* 2]  **Xingping Dong** [1]  **Yujia Wang** [3]  **Wenguan Wang** [2]

## Abstract

Despite rapid progress in generative 3D creation, producing high-fidelity 3D face assets from a single image remains challenging, as it requires both identity-critical facial micro-structures and fine-grained view-consistent textures. To address this, we present a two-stage framework named **SAMT** for monocular 3D avatar generation and texture synthesis. Specifically, a latent 3D diffusion model for facial mesh generation is pretrained and then further adapted to generate high-quality facial geometry through large-scale domain-specific finetuning on 35K curated 3D avatar models. Subsequently, the generated facial mesh is textured through a multi-view-aware texturing strategy. It incorporates multi-view facial priors along with the mesh geometry to guide a 2D texturing diffusion, enabling cross-view consistent and mesh-aligned texture synthesis. Extensive experiments demonstrate that SAMT improves over existing baselines by producing more coherent facial geometry together with more fine-grained and view-consistent textures. Project page is available at https://github.com/muyuWang/SAMT.

## 1. Introduction

Recent advances in 3D generative modeling (Liu et al., 2023; Long et al., 2024) have made single-image 3D asset generation increasingly feasible. Among diverse object categories, human faces are a particularly high-impact target for 3D generation due to their wide applications in fields such as AR/VR and digital entertainment. To fit the demands of

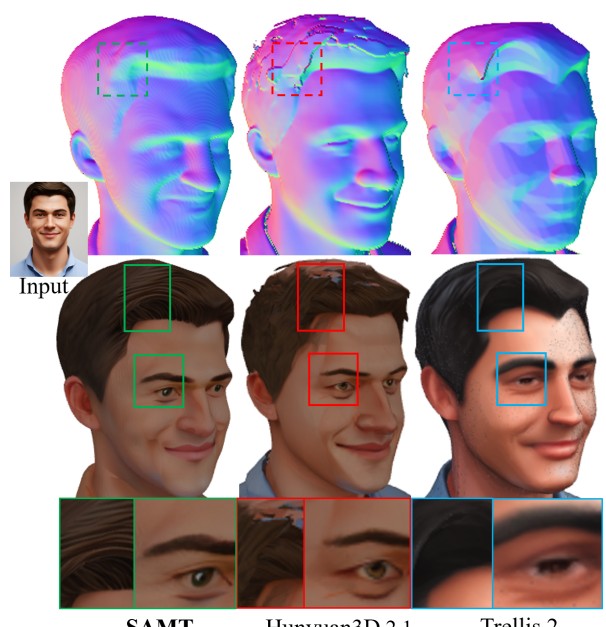

*Figure 1.* Textured 3D facial meshes generated by the complete pipelines of SAMT, Hunyuan3D 2.1 (Hunyuan3D et al., 2025) and Trellis 2 (Xiang et al., 2025a). Hunyuan3D 2.1 fails to generate regular hair geometry and view-consistent textures, while Trellis 2 produces coarse textures with hair colors inconsistent with the input. Additionally, they struggle to synthesize natural-looking eyes in textures. In contrast, SAMT yields well-structured facial geometry and refined textures consistent with the input image.

these applications, 3D face assets require identity-critical micro-structures and clean, fine-grained appearances which are difficult for generative models to capture. This makes single-image-to-3D-face generation highly challenging.

Existing single-image-to-3D-assets methods focus on mesh synthesis by adapting the latent-diffusion paradigm (Ho et al., 2020) from 2D to 3D. These approaches use latent vector sets (Zhao et al., 2023; Zhang et al., 2024b; Li et al., 2025a; Zhao et al., 2025; Hunyuan3D et al., 2025) or voxel-based latents (Ren et al., 2024; Xiang et al., 2025b; Wu et al., 2025; Xiang et al., 2025a) to represent 3D objects and can directly decode the latents into explicit meshes after diffusion. Trained on large-scale and category-diverse 3D datasets, they can generate high-quality meshes with rich surface detail. However, when performing facial generations, these approaches often suffer from inaccurate fa-

---

[*]Equal contribution  [1]School of Computer Science, National Engineering Research Center for Multimedia Software, Institute of Artificial Intelligence, Hubei Key Laboratory of Multimedia and Network Communication Engineering, Wuhan University, Wuhan, China [2]The State Key Lab of Brain-Machine Intelligence, Zhejiang University, Zhejiang, China [3]Zhejiang Sci-Tech University, Zhejiang, China. Correspondence to: Xingping Dong <xingpingdong@whu.edu.cn>.

*Proceedings of the 43rd International Conference on Machine Learning*, Seoul, South Korea. PMLR 306, 2026. Copyright 2026 by the author(s).

cial geometry and irregular mesh structures. Additionally, the generated meshes usually lack a textured appearance, and synthesizing realistic facial textures for them remains an open challenge. Current approaches typically perform texturing in UV space (Yu et al., 2024) or use 2D diffusion (Zhao et al., 2025) to produce multi-view projections. Yet, they struggle to ensure texture fidelity and cross-view consistency, especially for identity-sensitive details.

To address the limitations of existing methods, this work proposes **SAMT**, a two-stage framework that sequentially generates structured, detailed facial meshes and fine-grained, view-consistent textures from a single input image. Fig. 1 highlights the effectiveness of the proposed SAMT in generating both facial geometry and textures.

In the first stage of SAMT, given an input image, a native image-conditioned 3D diffusion model based on DiT architectures is applied to generate a geometrically detailed face mesh. In order to enhance the generated geometric fidelity, we equip the 3D diffusion model with a VAE augmented by sharp-edge sampling and dual cross-attention (Chen et al., 2025b). The 3D diffusion model is pretrained on large-scale generic 3D models and then finetuned on our collected facial dataset to specialize in structured facial generation. The collected dataset includes 35K high-quality face models covering both real captured meshes with corresponding photos and synthetic meshes with rendered images, which provide a large-scale, mixed-source domain prior that enables our model to robustly generate coherent facial meshes with geometry faithfully matching the input image. In the second stage of SAMT, the facial texture of the generated mesh is synthesized with the same input image. We propose a multi-view-aware texturing strategy and employ a pretrained geometry-conditioned 2D texturing diffusion model to generate multi-view projections that are baked into a unified texture map. Due to the complexity of facial geometry, this 2D diffusion model often produces multi-view textures with limited cross-view consistency. Therefore, we introduce explicit multi-view priors from a facial multi-view generation model (Lyu et al., 2025) by pairing them with the mesh geometry conditions to jointly guide the diffusion model. We also propose a mesh-multi-view-priors alignment method to better preserve the identity features and layout from the priors. These designs enable the texturing diffusion model to synthesize identity-preserving and view-consistent textures that are faithfully aligned with the mesh surface.

In summary, SAMT provides a solution for an under-explored task that remains difficult for existing methods: generating an explicit textured 3D facial asset from a single image. Extensive experiments conducted on a wide range of avatar images demonstrate that SAMT produces more structured facial geometry and more faithful facial textures than previous approaches. Ablation studies further verify

the effectiveness of each component in SAMT.

**Conflict of Interest Disclosure.** The authors declare no financial or other substantive conflicts of interest.

## 2. Related Work

**3D Generation.** 3D generation has been explored with diverse representations, including implicit fields (Mildenhall et al., 2021; Wang et al., 2025), point clouds (Yang et al., 2019), meshes (Nash et al., 2020; Li et al., 2025a), and 3D Gaussian representations (Kerbl et al., 2023; Gao et al., 2025; Chen & Wang, 2024; Gao et al., 2026; Feng et al., 2025; Liu et al., 2026). Early works achieve 3D generation with score distillation from pretrained 2D diffusion models (Poole et al., 2023; Wang et al., 2023b; Chen et al., 2023), but require costly per-sample optimization and often struggle with multi-view consistency and category generalization. To overcome these limitations, native 3D generative models have been developed to learn directly from 3D data with point clouds (Yang et al., 2019; Zhang et al., 2023), meshes (Nash et al., 2020), or implicit surfaces (Mescheder et al., 2019; Peng et al., 2020). Building on the success of diffusion models (Ho et al., 2020; Li et al., 2026; Chen et al., 2025a; Zhang et al., 2026), recent works conduct diffusion in 3D spaces and compress shapes into latent embeddings (Vahdat et al., 2022; Zhang et al., 2022; Zhao et al., 2023) to boost processing efficiency. Cutting-edge latent 3D diffusion frameworks (Zhao et al., 2023; Wu et al., 2025; Li et al., 2025a) achieve greater scalability, but still tend to produce smoothed geometry and limited fine details.

**3D Avatar Generation.** The 3D Morphable Model (Blanz & Vetter, 1999) offers explicit face control via linear shape bases and allows parametric controllable 3D face generation (Li et al., 2017; Booth et al., 2018). Subsequent models further enhance the realism of generated faces. Some 3D-aware GAN frameworks (Yuan et al., 2023; Bhattarai et al., 2024; Li et al., 2023; An et al., 2023) integrate volumetric neural representations with adversarial training to synthesize heads with improved fidelity. Recent Gaussian-based avatar creation methods (Qian et al., 2024; Zhai et al., 2025) further explore efficient animatable avatar modeling with explicit 3D Gaussian representations. Moreover, diffusion models have also been applied to 3D avatar generation. Several typical methods enhance diffusion models by integrating more 3D priors to generate high-quality and controllable 3D head avatars, such as volume representations (Wang et al., 2023a; Zhang et al., 2024a), morphable meshes (Chen et al., 2024), or data-driven head models (Lyu et al., 2025). However, many of these methods do not directly provide explicit 3D assets for consistent re-rendering.

**Mesh Texture Synthesis.** Current mesh texture synthesis approaches can be broadly categorized into several

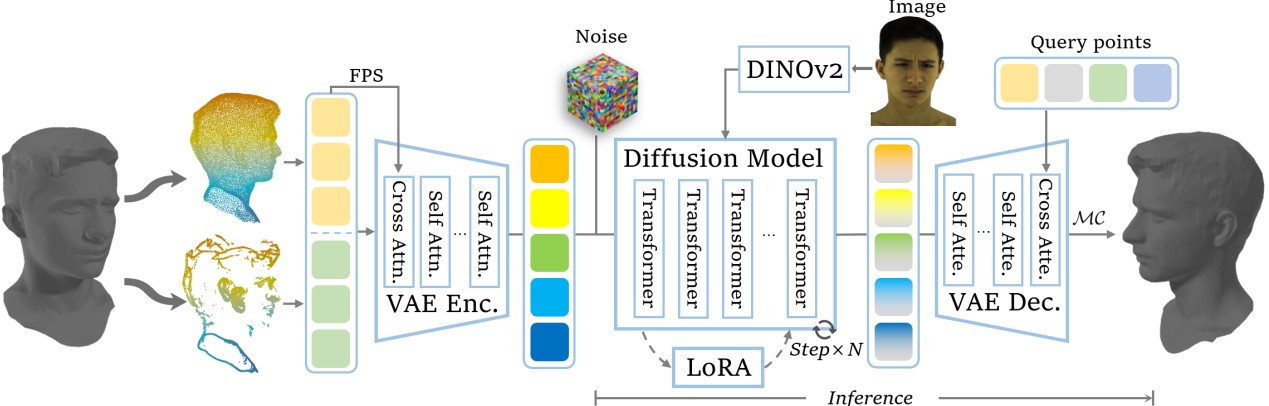

*Figure 2.* Architecture of our 3D diffusion model for facial mesh generation. During training, the DiT is pretrained and then jointly optimized with LoRA parameters on our collected large-scale facial dataset. At inference time, this model takes an input avatar image and generates a corresponding mesh through a feed-forward process. See § 3.1 for more details.

paradigms. The first paradigm generates textures in UV space (Cheng et al., 2024; Yu et al., 2023; 2024) and often performs poorly in generating textures for geometrically complex objects. A recent face-specific UV pipeline, Avatar-Tex (Qiu et al., 2026), achieves facial texture generation through diffusion-guided UV completion and GAN-based refinement, but it relies on fixed-topology meshes and places particular emphasis on stylized textures. Another paradigm synthesizes textures through geometry-conditioned generation and inpainting (Zhao et al., 2025; Cao et al., 2023; Zhang et al., 2024c; Richardson et al., 2023). These methods produce appealing texture details, but their fidelity is often constrained by limited multi-view consistency. Moreover, Trellis 2 (Xiang et al., 2025a) provides a new paradigm that directly synthesizes textures within its proposed 3D voxel space, which ensures multi-view consistency but often loses fine texture details. Specifically designed for facial textures, this work introduces a multi-view-aware texturing strategy to enforce view consistency and enhance texture details.

## 3. Method

SAMT aims to generate geometrically coherent meshes and multi-view-consistent textures from an input avatar image. This section introduces the two stages of SAMT in detail, targeting both mesh quality (§3.1) and texture fidelity (§3.2).

### 3.1. Stage 1: Face-specialized 3D Latent Diffusion

Given an input image of a human avatar $I$, the first stage of SAMT generates a structured facial mesh $M$ with fine-grained geometry. Inspired by recent 3D latent-set diffusion systems (Zhao et al., 2023; Li et al., 2025a; Zhao et al., 2025; Hunyuan3D et al., 2025), our geometry generation pipeline builds upon an image-conditioned 3D latent diffusion backbone. The diffusion process is performed in the latent space encoded by a 3D Shape Variational Auto-Encoder (VAE).

After denoising, the VAE decoder maps the latent code to an occupancy field, from which the generated mesh is obtained through Marching Cubes (Lorensen & Cline, 1998). Fig. 2 illustrates the network architecture and feed-forward inference process of our 3D diffusion model.

**3D Shape Variational Auto-Encoder.** 3DShape2VecSet (Zhang et al., 2023) proposes an effective variational autoencoder to encode 3D geometry into compact latent codes. However, its encoder relies on uniformly sampled surface points, which tends to under-represent high-frequency surface details. This makes it less suitable for modeling highly detailed facial geometry. Therefore, we adopt the subsequent DoraVAE (Chen et al., 2025b) as the VAE component of our 3D diffusion model. DoraVAE introduces Sharp Edge Sampling (SES) to better preserve fine geometry by augmenting uniform surface samples around geometrically salient edges, thereby allocating more reconstruction capacity to high-curvature regions. Moreover, it employs dual cross-attention to fuse global surface contexts and local details. It samples two point sets using uniform sampling (US) and sharp edge sampling, respectively, and separately computes their cross-attention features $\boldsymbol{C}_u, \boldsymbol{C}_s$:

$$\begin{aligned} \boldsymbol{C}_u &= \mathrm{CrossAttn}(\boldsymbol{P}_a, \boldsymbol{P}_u, \boldsymbol{P}_u), \\ \boldsymbol{C}_s &= \mathrm{CrossAttn}(\boldsymbol{P}_a, \boldsymbol{P}_s, \boldsymbol{P}_s), \end{aligned} \quad (1)$$

where $\boldsymbol{P}_u$ denotes uniformly sampled points, $\boldsymbol{P}_s$ denotes sharp-edge-sampled points and $\boldsymbol{P}_a$ is their fusion after farthest point sampling (FPS). The final point cloud feature $\boldsymbol{C}$ is the combination of both attention results: $\boldsymbol{C} = \boldsymbol{C}_u + \boldsymbol{C}_s$. This approach injects salient geometric cues into the latent representation, which is especially helpful for retaining facial micro-structures in downstream latent diffusion.

**3D Diffusion with Two-phase Training.** Following previous works (Li et al., 2025a; Zhao et al., 2025; Li et al., 2025b), our 3D generative model is also built on a DiT (Peebles & Xie, 2023) architecture. For the input image $I$, we

remove the background and center the foreground part. Afterwards, DINOv2 (Oquab et al., 2024) with an input size of $518 \times 518$ is applied as the image encoder to extract the image condition token sequence $\boldsymbol{c}$, which is injected into the latent feature through cross-attention. The diffusion network $\epsilon_\theta(\cdot)$ then models a conditional latent distribution and iteratively recovers a clean latent code $\boldsymbol{S}$ consistent with $\boldsymbol{c}$. This latent code is decoded into an implicit occupancy field, from which we extract the final mesh $M$:

$$\hat{o}(\boldsymbol{x}) = \mathcal{D}(\boldsymbol{S}, \boldsymbol{x}), \qquad M = \mathcal{MC}(\hat{o}), \qquad (2)$$

where $\mathcal{D}$ is our VAE decoder, $\boldsymbol{x} \in \mathbb{R}^3$ is a spatial query point, $\hat{o}(\cdot)$ is the final decoded occupancy field, and $\mathcal{MC}$ denotes Marching Cubes (Lorensen & Cline, 1998). Our diffusion network is trained with the noise-prediction objective:

$$\mathcal{L}_{\mathrm{diff}} = \mathbb{E}_{\boldsymbol{S}_0, t, \epsilon} \left[ \| \epsilon - \epsilon_\theta(\boldsymbol{S}_t, t, \boldsymbol{c}) \|_2^2 \right]. \qquad (3)$$

Here $\boldsymbol{S}_0$ is the clean latent code and $\boldsymbol{S}_t$ denotes the corresponding noisy latent at timestep $t$.

In order to generate geometrically detailed and structurally coherent facial meshes, we adopt a two-phase training strategy for our 3D generator. In the first phase, the denoiser network $\epsilon_\theta$ is trained on large-scale general 3D datasets to learn a strong, generic 3D prior and acquire 3D generation capability under single-image conditioning. In the second phase, we conduct training on a broad collection of 3D face assets to enhance performance and stability in fine-grained facial geometry modeling. We attach LoRA (Hu et al., 2022) adapters to the DiT backbone and optimize them jointly during this face-domain training. Concretely, for each linear transformation with weight matrix $W$, LoRA $\phi = \{A, B\}$ introduces a learnable low-rank update:

$$W' = W + \Delta W, \qquad \Delta W = BA, \qquad (4)$$

where $A \in \mathbb{R}^{r \times d_{\mathrm{in}}}$, $B \in \mathbb{R}^{d_{\mathrm{out}} \times r}$, $r$ is the LoRA rank and $r \ll \min(d_{\mathrm{in}}, d_{\mathrm{out}})$. $\Delta W$ is the weight update from LoRA and $W'$ is the adapted weight matrix after applying this update. We jointly optimize the diffusion parameters $\theta$ together with LoRA $\phi$ on our collected face assets:

$$(\theta^*, \phi^*) = \arg \min_{\theta, \phi} \mathbb{E}_{(I, M)} \left[ \mathcal{L}_{\mathrm{diff}}(\theta, \phi; I, M) \right]. \qquad (5)$$

$\theta^*$ and $\phi^*$ denote the optimized diffusion and LoRA parameters. This design injects a powerful face-domain adaptation capacity into the entire DiT, leading to more stable denoising and more reliable generation of structured facial meshes with geometry closely aligned to the input image.

### 3.2. Stage 2: Multi-view-consistent Texture Synthesis

With the textureless facial mesh $M$ generated by our 3D diffusion model, the second module of SAMT aims to synthesize a high-quality texture that is both mesh-aligned and

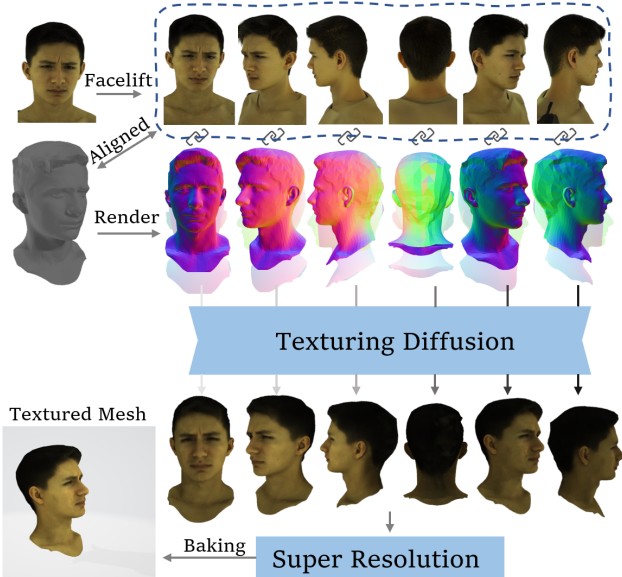

*Figure 3.* Illustration of our proposed facial-multi-view-aware texturing strategy. See § 3.2 for more details.

viewpoint-consistent. Fig. 3 presents our texture synthesis process for facial meshes. Following (Zhao et al., 2025; Hunyuan3D et al., 2025), we apply a pretrained geometry-conditioned 2D texturing diffusion model to generate view-space appearance maps guided by rendered geometric cues from $M$. Concretely, for a set of predefined camera poses $\{\boldsymbol{q}_v\}_{v=1}^V$, we render the geometry conditions $G_v$:

$$G_v = \mathcal{R}(M, \boldsymbol{q}_v), \qquad (6)$$

where $\mathcal{R}(\cdot)$ is the rendering function. $G_v$ includes geometry-derived normal maps $\boldsymbol{N}_v$ and canonical coordinate maps $\boldsymbol{K}_v$. The texturing diffusion then predicts a textured projection $X_v$ for each view from the input image $I$, with denoising guided by $G_v$. This ensures that the synthesized appearance conforms to the mesh surface. The texture-generating diffusion model is equipped with multi-view attention modules to achieve multi-view synthesis capability.

However, unlike highly symmetric generic objects, the single avatar input exhibits substantial geometric differences and limited symmetry with other viewpoints (e.g., side or rear head views). Consequently, directly applying the texturing diffusion model for facial inpainting often fails to maintain identity fidelity and consistency across different viewpoints. To address this, we propose a facial-multi-view-aware texturing strategy that introduces facial multi-view priors into the texturing process. Specifically, we leverage a multi-view face generator (Lyu et al., 2025) trained on large-scale face data, which is specialized for multi-view face synthesis and encodes strong face-domain 2D-to-3D priors. For the input avatar $I$, it generates $V = 6$ identity-consistent views $\{I_v\}_{v=1}^V$ at zero elevation with azimuths

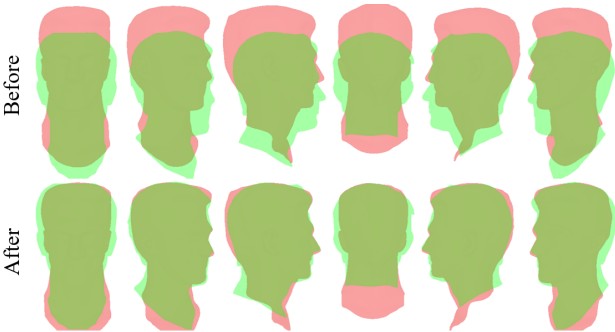

Before

After

*Figure 4.* The alignments between the generated mesh and the facial multi-view priors before and after applying our optimized transformation. Red contours indicate the projected silhouettes of the mesh, and green contours represent the foreground silhouette of the multi-view face images. See § 3.2 for more details.

in $\{0, \pm 45°, \pm 90°, 180°\}$. We align the mesh to these generated views under the corresponding camera poses, and inject these views into the texturing diffusion process to achieve high-quality, multi-view-consistent facial textures.

**Mesh-image Alignment.** In our experiments, we observe that the diffusion model produces textures that better match the input when the mesh projection is structurally aligned with the input image under the corresponding viewpoint. This is because the denoising process minimizes changes to the input image layout and local features in this case, preserving the original facial details. However, although the face-domain finetuning encourages our 3D diffusion model to generate facial meshes with a stable frontal view and the same canonical orientation as the facial priors, the generated mesh is still not guaranteed to be projection-aligned with the synthesized multi-view prior images $\{I_v\}_{v=1}^V$. Therefore, we introduce a derivative-free alignment optimization to address this mismatch without altering the mesh topology. Concretely, we solve for a global scale $s \in \mathbb{R}_+$ and translation $\boldsymbol{f} \in \mathbb{R}^3$ applied to the mesh, so that its rendered silhouettes $A_v^{\mathrm{mesh}}(\boldsymbol{s}, \boldsymbol{f})$ best match the foreground mask $A_v^{\mathrm{img}}$ of $\{I_v\}_{v=1}^V$ under their camera poses. We jointly optimize $(\boldsymbol{s}, \boldsymbol{f})$ by minimizing the mean silhouette mismatch across all views using an IoU-based objective:

$$(\boldsymbol{s}^\star, \boldsymbol{f}^\star) = \arg\min_{s,f} \frac{1}{V} \sum_{v=1}^V \left(1 - \mathrm{IoU}\big(A_v^{\mathrm{mesh}}(\boldsymbol{s}, \boldsymbol{f}),\, A_v^{\mathrm{img}}\big)\right). \tag{7}$$

Since this objective depends on non-differentiable rasterization, we solve it with the derivative-free Powell (Powell, 1964) method. We then apply the optimized transform $(\boldsymbol{s}^\star, \boldsymbol{f}^\star)$ to the mesh to align it with the multi-view facial priors, as shown in Fig. 4. This transformation helps improve the identity preservation and view consistency in the subsequent diffusion-based texturing.

**View-matched Texture Generation.** Due to the fixed structure of human heads, the six generated face views $\{I_v\}_{v=1}^V$

provide broad coverage of the visible head surface for texturing the aligned mesh. We explicitly incorporate the multi-view facial priors into the diffusion denoising process and design a view-conditioned one-to-one denoising scheme. As shown in Fig. 3, we feed the six facial views in one batch, each paired with the rendered geometric conditions $G_v$ from the corresponding viewpoint. These image-geometry pairs are individually used to synthesize textures from their respective viewpoints. With this view-matched strategy, the texturing diffusion receives not only well-defined view-consistency guidance, but also rich appearance detail from each view during texture generation. The generated multi-view texture projections are super-resolved and then baked into a single UV texture for the mesh. Together, these designs enable SAMT to synthesize mesh-aligned and multi-view-consistent textures from the input face image.

## 4. Experiments

In this section, training dataset and implementation details are introduced. We conduct qualitative and quantitative evaluations to validate the effectiveness of the proposed facial mesh and texture generation methods in SAMT. Additional video results are provided in the supplementary material.

### 4.1. Implementation

**Data Collection.** To enable our face generation model to learn rich face-domain geometric priors, we collect and curate a large set of high-quality 3D head meshes from three public datasets: Ava-256 (Martinez et al., 2024), FaceScape (Yang et al., 2020) and NPHM (Giebenhain et al., 2023). These datasets provide complementary high-quality 3D face assets with calibrated multi-view observations, covering diverse identities, rich expression variations, and fine-grained geometric detail. We further preprocess the collected meshes by cleaning artifacts, cropping away the torso region, and enforcing watertightness via mesh completion, followed by standardizing scale and coordinate conventions. Finally, we obtain 35K consistent watertight 3D face meshes spanning 1,300+ identities with 20+ expressions per identity, along with their multi-view observation images. These assets form the broad training set for our 3D diffusion model, covering diverse skin tones, hairstyles, and expressions.

**Training Setting of 3D Generation.** During training, the processed mesh is sampled to 16,384 points through uniform sampling and sharp edge sampling, separately, to fit in the VAE encoder. The VAE then encodes the geometry into a set of 2,048 latent tokens with an embedding dimension of 64. First, the 3D diffusion is trained on the Objaverse-XL (Deitke et al., 2023) dataset with a shared approach in the 3D generation field (Liu et al., 2024; Li et al., 2025a; Zhao et al., 2025; Hunyuan3D et al., 2025). Afterward, we finetune it on our collected facial dataset to enhance

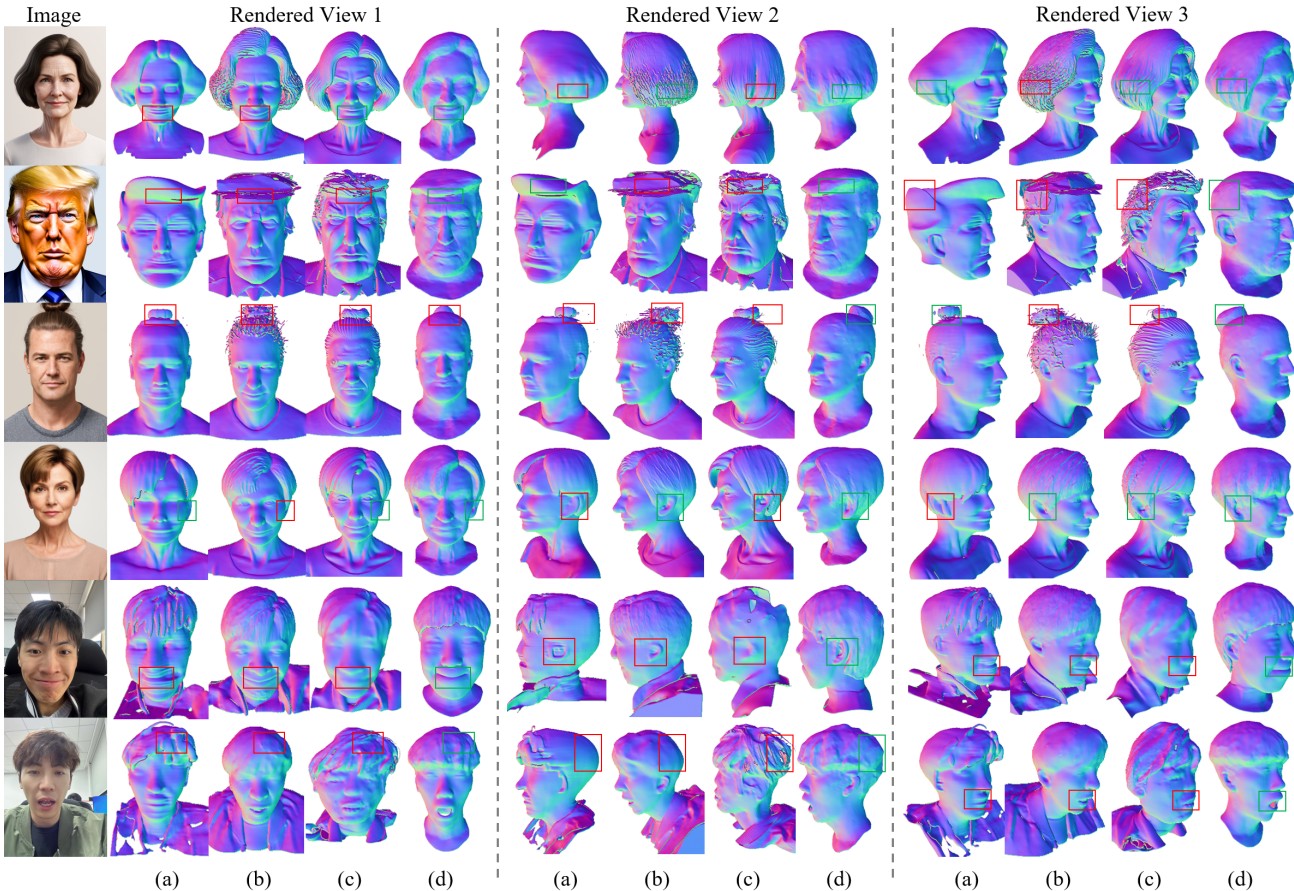

*Figure 5.* **Qualitative comparisons** in terms of 3D Facial Generation. Normal maps of facial meshes generated by (a) CraftsMan3D (Li et al., 2025a), (b) Hunyuan3D 2.1 (Hunyuan3D et al., 2025), (c) Trellis 2 (Xiang et al., 2025a), and (d) SAMT are rendered. SAMT can generate structured face geometry that is more consistent with the input image. Red rectangles highlight geometric distortions, while green rectangles indicate well-structured results. See § 4.2 for more details.

*Table 1.* **Quantitative comparisons** in terms of facial mesh generation (§ 4.2). Best results are highlighted as **1st** and 2nd.

| Models | ULIP-I ↑ | ULIP-T ↑ | Uni3D-I ↑ | Uni3D-T ↑ | CD ↓ | F1 ↑ |
|---|---|---|---|---|---|---|
| CraftsMan3D | 0.131 | 0.061 | 0.284 | 0.223 | 0.123 | 0.166 |
| Hunyuan3D 2.1 | 0.147 | 0.072 | 0.325 | 0.243 | 0.121 | 0.171 |
| Trellis 2 | 0.140 | 0.070 | **0.327** | 0.239 | 0.116 | 0.183 |
| SAMT | **0.149** | **0.084** | 0.323 | **0.251** | **0.105** | **0.216** |

structured geometry generation and improve fidelity in the face domain. We insert LoRA with rank $r = 32$ into the DiT backbone and jointly optimize the backbone parameters and LoRA adapters. A grouped learning-rate strategy for the AdamW optimizer is applied: $5 \times 10^{-6}$ for the backbone and $2 \times 10^{-4}$ for LoRA, with a cosine schedule.

### 4.2. Evaluation of 3D Facial Generation

We first evaluate the effectiveness of our face-specialized 3D latent diffusion model on the task of single-image facial mesh generation. Although several avatar-specific methods (An et al., 2023; Cha et al., 2025; Gerogiannis

et al., 2025; Lyu et al., 2025) generate high-quality neural-rendered or parametric avatars, they often do not directly output explicit meshes or require identity-specific optimization, making them less directly comparable under our feed-forward mesh generation protocol. Therefore, we qualitatively and quantitatively compare SAMT against representative state-of-the-art mesh generation methods including CraftsMan3D (Li et al., 2025a), Hunyuan3D 2.1 (Hunyuan3D et al., 2025), and Trellis 2 (Xiang et al., 2025a) under the same single-image-to-explicit-3D-asset setting.

In Fig. 5, we present qualitative comparisons on a diverse set of portrait inputs covering multiple styles, including studio-like images, in-the-wild photos and stylized portraits. CraftsMan3D often produces noticeably over-smoothed and distorted head shapes, leading to weakened identity cues and blurred structures. Hunyuan3D 2.1 and Trellis 2, despite being trained on substantially larger and more diverse 3D corpora, exhibit reduced robustness on head generation. They may generate overly sharp, spiky hair geometry and often suffer from distortions when applied to real-world

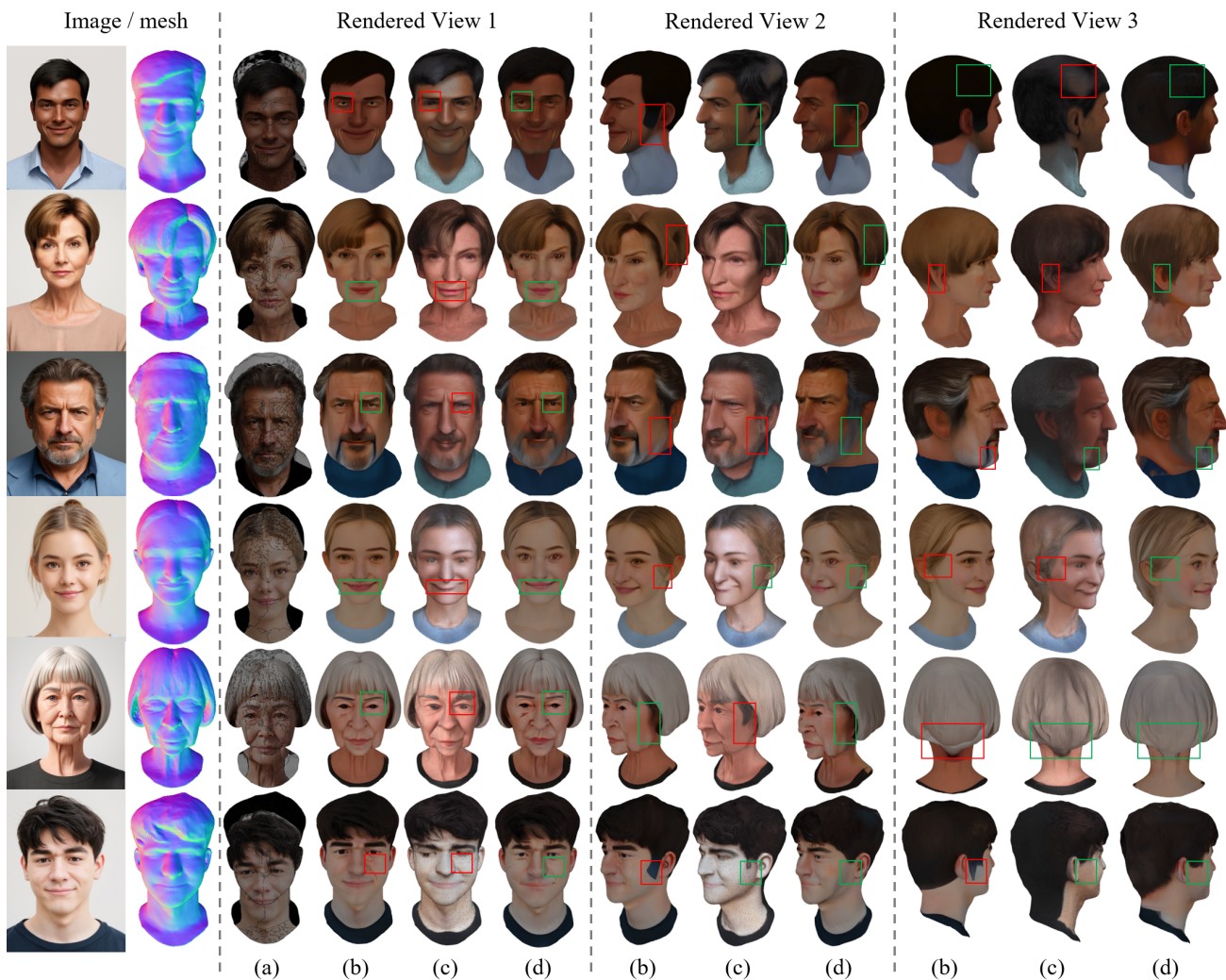

| Image / mesh | Rendered View 1 | Rendered View 2 | Rendered View 3 |
| --- | --- | --- | --- |
| | (a) (b) (c) (d) | (b) (c) (d) | (b) (c) (d) |

*Figure 6.* **Qualitative comparisons** in terms of facial texture generation. (a), (b) and (c) respectively stand for the baseline method TEXGen (Yu et al., 2024), Hunyuan3D 2.1 (Hunyuan3D et al., 2025) and Trellis 2 (Xiang et al., 2025a). (d) represents our SAMT. Blurry or inconsistent textures are marked in red, while view-consistent ones are marked in green. Please zoom in to better visualize texture details such as the hair, eyes and wrinkles. See § 4.3 for more details.

photos. Moreover, their generated heads tend to exhibit an unnatural forward tilt and struggle to produce structurally plausible geometry. In contrast, SAMT can generate facial meshes that better match the input identity while retaining structured geometry. It more reliably completes regions that are not directly observed in the input image, such as the side face profile. Moreover, as shown in the bottom two rows of Fig. 5, SAMT is more robust to variations in image style, especially in real captured images where prior methods often produce unstable geometry and noticeable distortions.

Table 1 presents quantitative comparisons across different methods. For evaluation without ground truth, we apply ULIP (Xue et al., 2023) and Uni3D (Zhou et al., 2024) to measure the cross-modal similarity between the generated 3D face and the input image. Specifically, for each input

*Table 2.* **Quantitative comparisons** of different methods in terms of facial texture generation (§ 4.3).

| Models | CLIP-score ↑ | FID-CLIP ↓ | CMMD ↓ | ArcFace ↓ |
| --- | --- | --- | --- | --- |
| TEXGen | 0.697 | 60.308 | 5.213 | 0.442 |
| Hunyuan3D 2.1 | 0.821 | 33.401 | 2.976 | 0.329 |
| Trellis 2 | 0.814 | 36.784 | 3.271 | 0.388 |
| SAMT | **0.839** | **32.211** | **2.805** | **0.316** |

image, GPT-5.2 is used to produce a detailed textual description and a facial mesh is generated from the image. Afterwards, the meshes are sampled to a point-cloud representation for evaluation. Results are averaged over 50 test cases and reported as ULIP-I/Uni3D-I for point-cloud-image similarity and ULIP-T/Uni3D-T for point-cloud-text similarity. For evaluation with ground-truth, we use samples from the RenderMe-360 (Pan et al., 2023) dataset to

measure the geometric accuracy of generated facial meshes using Chamfer Distance (CD) and F1 score with a threshold of 0.01. As shown in Table 1, SAMT yields the best performance on most metrics, indicating that it achieves better consistency with the input facial image and higher geometric accuracy compared with existing approaches.

## 4.3. Evaluation of Facial Texture Synthesis

Building on the generated high-fidelity facial meshes, we further evaluate the texture synthesis quality of SAMT. We compare SAMT against three state-of-the-art baselines including Hunyuan3D 2.1 (Hunyuan3D et al., 2025), Trellis 2 (Xiang et al., 2025a), and TEXGen (Yu et al., 2024). Given the same untextured mesh and the input face image, we apply each baseline and SAMT to synthesize textures. The resulting qualitative comparisons are shown in Fig. 6. TEX-Gen performs facial texture synthesis directly in UV space, but fails to align the generated textures with the complex facial geometry, thereby degrading visual quality. Since the input image only provides a frontal view, Hunyuan3D 2.1 cannot reliably enforce appearance consistency between the front view and the unseen side and back regions. Trellis 2 generates textures directly in the 3D O-Voxel representation, which encourages view-consistent appearance, but often produces mismatched details between the input image and generated texture. In contrast, benefiting from the introduced multi-view facial priors and our mesh-image alignment strategy, SAMT synthesizes multi-view-consistent facial textures while preserving identity-specific features and maintaining strong shape-image alignment.

For quantitative evaluation, we use image-level metrics to measure the consistency between the input images and renderings of the generated textures, including CLIP-score (Radford et al., 2021) for evaluating semantic alignment, FID-CLIP (Parmar et al., 2022) for measuring the distribution distance in CLIP feature space, and CMMD (Jayasumana et al., 2024) for capturing CLIP-based distribution discrepancy with higher sensitivity to fine-grained details. Additionally, for the face generation task, we include the ArcFace (Deng et al., 2019) metric implemented by Deep-Face (Serengil & Ozpinar, 2021) to quantify identity preservation in the rendered appearances. SAMT achieves consistently better quantitative results than baselines in Table 2, revealing that it produces textures with higher rendering quality and better preservation of identity cues.

## 4.4. Ablation Study

We further conduct comprehensive ablation studies to analyze the impact of each key component in our facial mesh generation and texture synthesis workflow.

**Face Domain Adaptation.** In the second training phase of our 3D diffusion model, it is finetuned on our collected

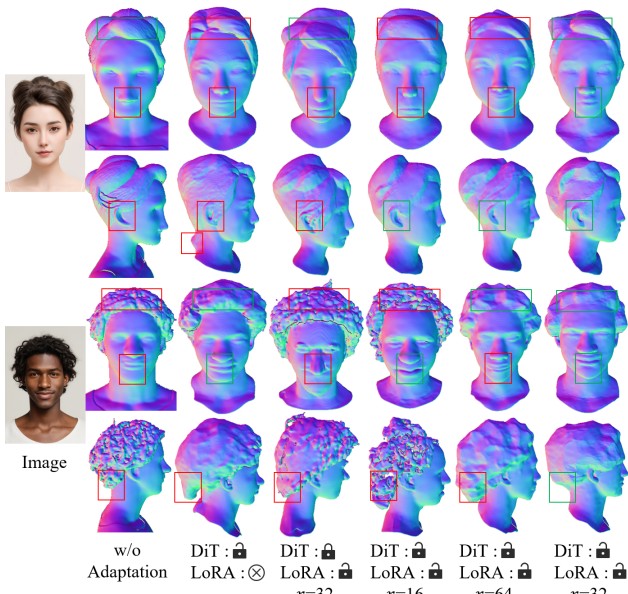

*Figure 7.* **Qualitative ablation results** of the face-domain adaptation and different adaptation strategies for the 3D facial generation model. $r$ indicates the rank of LoRA. See § 4.4 for more details.

*Table 3.* **Quantitative ablation results** of the face-domain adaptation and different adaptation strategies (§ 4.4).

| Models | ULIP-I ↑ | ULIP-T ↑ | Uni3D-I ↑ | Uni3D-T ↑ | CD ↓ |
|---|---|---|---|---|---|
| w/o Adaptation | 0.133 | 0.065 | 0.291 | 0.229 | 0.120 |
| DiT unfrozen, w/o LoRA | 0.138 | 0.076 | 0.316 | 0.236 | 0.110 |
| DiT frozen, LoRA $r = 32$ | 0.135 | 0.075 | 0.309 | 0.233 | 0.116 |
| DiT unfrozen, LoRA $r = 16$ | 0.146 | 0.081 | 0.316 | 0.247 | 0.115 |
| DiT unfrozen, LoRA $r = 64$ | 0.143 | 0.079 | 0.320 | 0.245 | 0.108 |
| DiT unfrozen, LoRA $r = 32$ | **0.149** | **0.084** | **0.323** | **0.251** | **0.105** |

face dataset. We ablate the face-domain adaptation in the second training phase and different adaptation strategies for the 3D diffusion model, including finetuning the DiT without LoRA, LoRA-only adaptation with the DiT frozen, and jointly training LoRA and the DiT with different LoRA ranks $r \in \{16, 32, 64\}$. As shown in Fig. 7, without finetuning on our collected face dataset, the 3D diffusion model produces over-smoothed facial geometry and structural distortions, highlighting the necessity of face-domain adaptation for reliable facial geometry generation. DiT-only training often lacks the capacity to correct structural errors, while LoRA-only tuning provides limited adaptation and often leaves geometric distortions. Under joint training settings, rank 16 tends to under-adapt and miss fine details, while rank 64 may over-adapt and introduce artifacts. Table 3 presents quantitative ablation results for different adaptation strategies. Therefore, we use $r = 32$ in our joint training of DiT and LoRA, as this configuration yields the most stable facial structure with the highest geometric fidelity.

**Mesh-image Alignment.** To ablate the mesh-image alignment strategy, we directly use the mesh generated by the 3D diffusion model for texture synthesis without applying

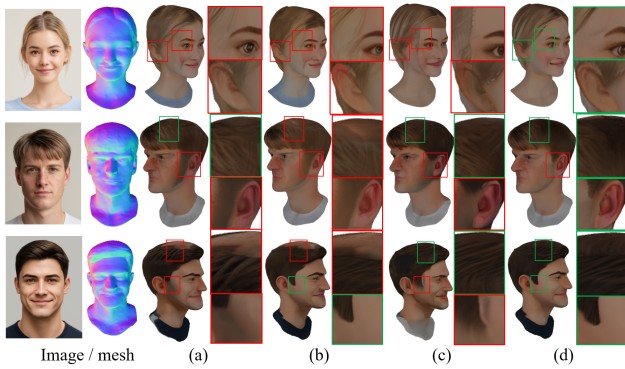

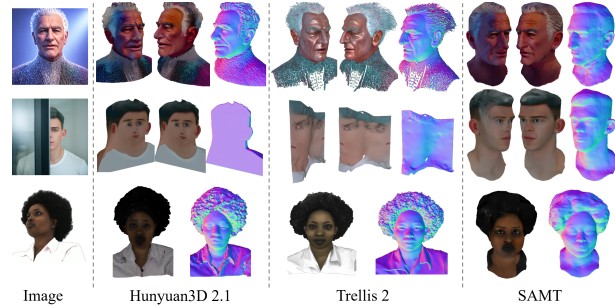

Image / mesh      (a)      (b)      (c)      (d)

*Figure 8.* **Qualitative ablation results** of the mesh-image alignment and facial multi-view prior in texture synthesis. (a), (b), (c), and (d) respectively denote the foundation texturing diffusion, w/ facial multi-view prior, w/ mesh-image alignment, and the full texture synthesis workflow of SAMT. See § 4.4 for more details.

*Table 4.* **Quantitative ablation results** of the components in the texturing process of SAMT (§ 4.4).

| Models | CLIP-score ↑ | FID-CLIP ↓ | CMMD ↓ | ArcFace ↓ |
|---|---|---|---|---|
| Foundation texturing diffusion | 0.823 | 33.410 | 2.985 | 0.329 |
| w/ facial multi-view prior | 0.833 | 32.406 | 2.821 | 0.319 |
| w/ mesh-image alignment | 0.828 | 33.173 | 2.980 | 0.324 |
| SAMT | **0.839** | **32.211** | **2.805** | **0.316** |

Image      Hunyuan3D 2.1      Trellis 2      SAMT

*Figure 9.* **Visualizations of out-of-distribution cases**, including strong directional lighting (1st row), heavy occlusion (2nd row), and extreme profile view (3rd row). See § 4.5 for more details.

alignment. Results in Fig. 8 and Table 4 demonstrate that our mesh-image alignment enables the texture generation network to better preserve facial characteristics, improve view consistency and reduce misalignment artifacts.

**Facial Multi-view Prior.** We also evaluate texture synthesis with and without the incorporation of the multi-view facial priors in Fig. 8 and Table 4. Without these priors, the texturing diffusion model may fail to maintain appearance consistency across viewpoints and produce non-frontal textures that deviate from the frontal reference. Under this setting, generated textures in non-frontal views may also be over-smoothed and lack fine-grained details, such as the hair and sideburns in Fig. 8. Compared to the foundation texturing diffusion model, the proposed facial-multi-view-aware texturing strategy in SAMT not only improves multi-view consistency but also enhances texture fidelity.

### 4.5. Failure Cases and Limitations

Although SAMT improves the geometric fidelity and multi-view texture consistency of the generated facial assets, it still faces challenges in out-of-distribution cases such as strong directional lighting, heavy occlusions and extreme profile views. As shown in Fig. 9, heavy occlusions may degrade geometry fidelity and texture consistency, strong directional lighting may be baked into the synthesized texture, and extreme profile views may introduce ambiguity in unobserved facial regions. These issues are not unique to SAMT but reflect broader limitations in the task of single-image 3D

facial asset generation, where unseen geometry and appearance have to be inferred from limited visual evidence. Even in these cases, SAMT still produces more reliable facial geometry and more coherent textures than existing approaches, benefiting from the proposed face-domain adaptation and facial-multi-view-aware texturing strategy.

## 5. Conclusion

This paper proposes SAMT, a two-stage framework that sequentially generates facial meshes and textures from a single input image. A pretrained 3D diffusion model finetuned on our collected large-scale facial dataset is applied to generate a fine-grained and structured facial mesh, and the texture is synthesized by our proposed facial-multi-view-aware texturing strategy that ensures view consistency and identity preservation. SAMT facilitates the creation of practical 3D avatar assets with coherent geometry and faithful appearance. Extensive experiments validate the effectiveness and advantages of our SAMT for 3D facial asset generation.

## Impact Statement

This paper presents work whose goal is to advance the field of Machine Learning. There are many potential societal consequences of our work, none which we feel must be specifically highlighted here.

## Acknowledgements

This work was supported by the New Generation Artificial Intelligence National Science and Technology Major Project (No. 2025ZD0123501), the Fundamental Research Funds for the Central Universities (No. 2042026kf0044), the National Natural Science Foundation of China (No. 62471342, 62302046), Zhejiang Provincial Natural Science Foundation of China (No. LD25F020001), the CCF-Tencent Open Fund, the New Cornerstone Science Foundation through the XPLORER PRIZE, the Innovative Research Group Project of Hubei Province under Grant 2024AFA017, and WHU-Kingsoft Joint Lab.

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
