# OpenReview forum: "SAMT: Generating Structured Avatar Meshes and Textures from a Single Image"
_ICML.cc/2026/Conference — ICML 2026 regular_

### Official Review · Reviewer_EhpM · 2026-03-03

**Soundness:** 3
**Presentation:** 3
**Significance:** 2
**Originality:** 3
**Overall Recommendation:** 4
**Confidence:** 5

**Summary:**

This paper proposes SAMT, a two-stage pipeline that generates structured 3D face meshes and multi-view-consistent textures from a single image by fine-tuning a latent 3D diffusion model on 35K head meshes and then performing geometry-conditioned texture diffusion guided by synthesized multi-view face priors plus silhouette-based alignment. Experiments demonstrate the performance gain in geometry and texture compared to general 3D generation models. Ablation studies also evaluate the necessity of mesh-image alignment and facial multi-view prior module for generating view-consistent texture.

**Compliance With Llm Reviewing Policy:**

Affirmed.

**Final Justification:**

The authors' rebuttal have addressed my concerns.

**Key Questions For Authors:**

1. What's the main advantage compared to other similar methods? For example, single image to 3D face reconstruction methods, single image to 3d face generation (optimization-based methods using SDS loss, or other feed-forward methods like SAMT). All these relevant methods are not discussed in the related work part.

2. Failure modes and out-of-distribution behavior: How does SAMT behave on heavy occlusions, extreme hairstyles?

**Limitations:**

The results seem cartoonish compared to the real scan.

**Strengths And Weaknesses:**

Strengths

1. The paper is well written and easy to follow.
2. The paper combines three datasets and preprocesses to obtain 35k 3D face meshes, which is a useful dataset for future research.
3. Two-phase training with LoRA adaptation to improve geometry fidelity in the face domain and mesh-image alignment, as well as facial multi-view prior to enhance consistency in texture generation, is a reasonable and effective way. Experiments also demonstrate the effectiveness of the current method.

Weaknesses

1. Novelty is somewhat incremental: several core components appear adopted from existing lines (DiT latent 3D diffusion, DoraVAE, LoRA fine-tuning, geometry-conditioned 2D texturing diffusion). The whole method is more like an "engineering aggregation" rather than a distinctly new learning principle.
2. Comparison with other single-image face generation methods is missing. The compared baseline methods, such as Trellis2 and Hunyuan, are all methods developed for general object generation.
3. It would be better to add the model trained during training phase one without the 3D face mesh finetune to the ablation study to emphasize the importance of the finetune strategy.
4. Quantitative results are missing for the texture consistency improvement ablation. Only qualitative results are shown in Fig 8.

---

> ### Author Rebuttal · Authors · 2026-03-31
>
> We sincerely thank the reviewer for the constructive feedback, and respond to each point below.
>
> **W1**: We respectfully clarify that the goal of SAMT is **not** to introduce a new learning principle, but to tackle a practically important task that remains insufficiently addressed by existing methods: generating an explicit textured 3D facial asset from a single image. In this setting, existing methods often struggle to preserve facial structure and appearance consistency (Line 45-53). SAMT tackles this problem through a face-oriented mesh generation pipeline built on our curated facial dataset and a dedicated facial-multi-view-aware texturing strategy. The novelty of SAMT lies in showing that this task is not well addressed by directly aggregating existing components, and in providing an effective solution for single-image 3D facial asset generation through dedicated face-specific designs. This is supported by the comparisons in Sec 4.2 and 4.3, and ablations in Sec 4.4 together with W3 and W4. We therefore believe that SAMT should be viewed as more than an engineering aggregation.
>
> **W2**: We apologize for the confusion. Since SAMT targets explicit textured 3D facial asset generation from a single image, we mainly compared it with recent pipelines that also directly output explicit 3D meshes and textures. Many face-specific generation methods improve realism or controllability, but often do not provide explicit 3D assets (Line 85-88), therefore are not always directly comparable to SAMT due to differences in task setting and output representation. We will clarify this distinction more explicitly in the revision.
>
> **W3**: Thanks for your suggestion. We add ablations using 3D diffusion model without finetuning on our curated face mesh dataset in the following table, and qualitative results are shown in Figure 3 in the anonymous link https://anonymous.4open.science/r/anonymous-24E6, which will be added to Figure 7 in the revised paper. The results show that, without the face-domain adaptation, the model produces facial geometry that is less faithful to the input. We will add this ablation in the revision.
>   | Models                  | ULIP-I↑ | ULIP-T↑ | Uni3D-I↑ | Uni3D-T↑ | CD↓    |
> |-------------------------|-------:|-------:|--------:|--------:|------:|
> | w/o Adaptation          | 0.133  | 0.065  | 0.291   | 0.229   | 0.120 |
> | SAMT |**0.149**  | **0.084**  | **0.323**   | **0.251**   | **0.105** |
>
> **W4**: Thanks for pointing this out. The following table provides quantitative results of ablations for our proposed texturing method. Results indicate that both facial multi-view prior injection and mesh-image alignment contribute to the gain of SAMT. We will add this in the revision.
>   |                      | CLIP-score↑ | FID-CLIP↓ | CMMD↓  | ArcFace↓ |
> |---------------------------|-----------:|---------:|------:|--------:|
> | Foundation texturing diffusion | 0.823 | 33.410   | 2.985 | 0.329|
> | w. facial multi-view prior| 0.833 | 32.406 | 2.821 | 0.319|
> | w. mesh-image alignment | 0.828      | 33.173   | 2.980 | 0.324|
> | SAMT |**0.839**|**32.211**| **2.805** | **0.316** |
>
> **Q1**: Our main advantage is that SAMT is able to generate high-fidelity explicit textured 3D facial assets. Compared with 3D face reconstruction methods, SAMT targets asset generation rather than only geometric or parametric recovery, and directly outputs a high-quality textured facial mesh, which facilitates downstream asset rendering and processing. Compared with optimization-based methods using SDS, SAMT avoids expensive per-instance optimization and the instability of SDS. Instead, SAMT uses a practical two-stage feed-forward pipeline for geometry and texture synthesis, resulting in more practical generation time. Compared with other feed-forward methods, SAMT injects strong face-specific priors in face-domain training and texturing, which improves facial geometry structure and texture consistency. We will expand the related work accordingly and discuss more relevant works.
>
> **Q2**: SAMT shows a certain degree of robustness to hairstyle, as illustrated in the top two rows of Figure 2 in the anonymous link. Heavy occlusions remain more challenging, which is not specific to SAMT, but a broader difficulty for single-image 3D facial asset generation, as shown in the 2nd and 3rd row of Figure 1 in the link. Even in such cases, SAMT relatively better maintains a more stable overall facial structure than existing approaches. We will add the shown failure and out-of-distribution cases in the revision.
>
> **Limitations**: Our current pipeline prioritizes structural geometry and cross-view consistent texture under single-view reference. So appearance in unseen or weakly constrained regions may still look less realistic than a real scan, especially for out-of-distribution inputs. We also view improving realism under challenging conditions as an important direction for future work, and will discuss this in the revision.

---

> > ### Author Rebuttal · Reviewer_EhpM · 2026-04-04
> >
> > Thanks for your reply. For W2, I refer to methods such as FaceLift and PERSE, which utilize 3DGS as their representation. Point clouds can be extracted from 3DGS and utilized to compute the metrics adopted in this paper. Can you explain the advantages of your method compared to these methods, since they are a popular stream in the single image to 3D head generation area.

---

> > > ### Author Response · Authors · 2026-04-07
> > >
> > > We sincerely appreciate your constructive feedback and are glad that some concerns have now been addressed.
> > >
> > > ---
> > >
> > > We are sorry for the confusion in our previous response. SAMT has two practical advantages over 3DGS-based methods. 1) SAMT **achieves stronger geometric fidelity** with consistent textures (Line 29-31). Many 3DGS-based single-image head generation methods [1,2,3] are primarily designed for high-quality rendering and may be less reliable in terms of geometric quality, as reflected in the table below. Meanwhile, obtaining high-quality surfaces from their unorganized 3D Gaussians remains non-trivial [4]. 2) SAMT **directly produces more structured facial meshes** (Line 93-95), which are more suitable for downstream graphics workflows such as animation and gaming [5]. By contrast, deriving meshes for 3DGS-based avatar generation methods requires additional extraction and often results in noisier and less regular surfaces, as shown in Figure 4 of https://anonymous.4open.science/r/anonymous-24E6.
> > >
> > > In addition, we conducted experiments by extracting point clouds from the 3DGS representations and evaluating them with the same metrics as in our paper. Besides the mentioned FaceLift [1] and PERSE [2], we also include results for another recent 3DGS-based single-image 3D avatar generation method, Arc2Avatar [3]. As shown by the results in the table below, SAMT provides stronger geometric fidelity, especially on the ground-truth geometry evaluations (CD and F1). We also provide extra visual comparisons in Figure 4 of https://anonymous.4open.science/r/anonymous-24E6, where we extract meshes for the 3DGS-based methods by applying Marching Cubes to the 3DGS outputs. Meshes extracted from 3DGS-based methods exhibit noisy and irregular surfaces, while meshes generated by SAMT are cleaner, more structured, and more coherent in facial geometry.
> > >
> > > | Method | ULIP-I ↑ | ULIP-T ↑ | Uni3D-I ↑ | Uni3D-T ↑ | CD ↓ | F1 ↑ |
> > > |---|---:|---:|---:|---:|---:|---:|
> > > | FaceLift | 0.138 | 0.077 | 0.319 | 0.240 | 0.138 | 0.149 |
> > > | PERSE | 0.134 | 0.076 | 0.311 | 0.237 | 0.127 | 0.179 |
> > > | Arc2Avatar | 0.131 | 0.079 | 0.307 | 0.244 | 0.123 | 0.178 |
> > > | SAMT | **0.149** | **0.084** | **0.323** | **0.251** | **0.105** | **0.216** |
> > >
> > >
> > >
> > > [1] FaceLift: Learning Generalizable Single Image 3D Face Reconstruction from Synthetic Heads (ICCV 2025)
> > >
> > > [2] PERSE: Personalized 3D Generative Avatars from A Single Portrait (CVPR 2025)
> > >
> > > [3] Arc2Avatar: Generating Expressive 3D Avatars from a Single Image via ID Guidance (CVPR 2025)
> > >
> > > [4] SuGaR: Surface-Aligned Gaussian Splatting for Efficient 3D Mesh Reconstruction and High-Quality Mesh Rendering (CVPR 2024)
> > >
> > > [5] MaGS: Reconstructing and Simulating Dynamic 3D Objects with Mesh-adsorbed Gaussian Splatting (ICCV 2025)
> > >
> > > ---
> > >
> > > We thank the reviewer again for raising this point, and your comments greatly help improve the quality of our work. We will incorporate all the additional discussions, experiments, and visual results into the appropriate parts of the final version.

---

### Official Review · Reviewer_5j24 · 2026-03-12

**Soundness:** 3
**Presentation:** 4
**Significance:** 2
**Originality:** 2
**Overall Recommendation:** 4
**Confidence:** 3

**Summary:**

SAMT is a two-stage framework for monocular 3D avatar generation and texture synthesis. In the first stage, a face-specialized 3D latent diffusion model generates a structured facial mesh from a single image. In the second stage, a multi-view-aware diffusion-based texturing strategy produces detailed and cross-view consistent textures using facial multi-view priors. Experiments show that SAMT generates more realistic facial geometry, preserves identity information, and produces more view-consistent textures compared to existing methods.

**Compliance With Llm Reviewing Policy:**

Affirmed.

**Final Justification:**

Given the thorough rebuttal  I am raising my score to 4

**Key Questions For Authors:**

1. What is the average runtime for the full pipeline during inference, including mesh generation, alignment optimization, and texture synthesis?
2. How robust is the method to challenging real-world inputs, such as faces with accessories (e.g., glasses or hats)?
3. Could the authors clarify the claim of generating "topology-regular" and "production-grade" meshes via Marching Cubes? Is there a post-processing retopology step that was omitted from the text? If not, I strongly suggest toning down the claims regarding production-ready topology.

**Limitations:**

The paper lacks a dedicated discussion or visual demonstration of failure cases. The authors should add a paragraph and visual examples showing how the pipeline handles occlusions, more extreme profile views or strong directional lighting.

**Strengths And Weaknesses:**

Strengths:

1. The paper presents a clear and well-structured pipeline for single-image 3D avatar generation.
2. The use of multi-view facial priors for texture synthesis is a good idea. By conditioning the texture generation on several synthesized views of the face, the method helps maintain consistent appearance across different viewpoints, which is a common problem in single-image 3D reconstruction.
3. The paper includes ablation studies analyzing the influence of LoRA adaptation, mesh-image alignment, and the multi-view prior.
4. The qualitative results appear convincing. Compared to existing approaches, SAMT produces more structured facial geometry and more consistent textures, particularly in challenging regions such as hair, eyes, and side views.

Weaknesses:
1. The overall level of methodological novelty is somewhat limited. Many components of the system rely on existing techniques such as latent 3D diffusion, DiT, LoRA fine-tuning, geometry-conditioned diffusion, and multi-view rendering. The main contribution appears to be the integration and adaptation of these components rather than a fundamentally new modeling approach.
2. The method relies heavily on multiple external pretrained models, including DINOv2 for image encoding, a pretrained multi-view face generator, and a geometry-conditioned texturing diffusion model. It is therefore somewhat unclear how much of the final performance improvement is due to the proposed contributions versus the strength of these pretrained components.
3. The paper does not provide a detailed analysis of computational cost. It would be useful to know the training and inference time as well as hardware requirements.
4. The authors repeatedly claim that SAMT generates "topology-regular facial meshes" that are "production-grade 3D face assets" (e.g., lines 76, 133, 139). However, in line 146, it is stated that the final mesh is extracted using the Marching Cubes algorithm. Marching Cubes is well-known for producing irregular triangle soup, which is the exact opposite of regular, production-ready topology (which typically requires quad-based retopology). This terminology is highly misleading unless an unmentioned retopology step is performed.
The paper lacks a dedicated discussion of technical limitations and failure cases (e.g., extreme lighting, facial hair, heavy occlusions).

---

> ### Author Rebuttal · Authors · 2026-03-31
>
> We thank the reviewer for the careful reading and constructive comments, and respond to each point below.
>
> **W1**: We would like to clarify that SAMT is **not** intended as a fundamentally new modeling approach. Rather, its novelty lies in providing an effective face-specific solution for an under-explored task that remains difficult for existing methods: generating an explicit textured 3D facial asset from a single image. Existing methods often produce inaccurate facial geometry and inconsistent textures (Line 45- 53). In contrast, SAMT addresses this problem through a face-specific geometry generation pipeline built on our curated 35K facial dataset and a facial-multi-view-aware texturing approach. Our novelty is to show that the mentioned techniques are not sufficient on their own for single-image 3D facial asset generation. Instead, it is our face-specific system design that enables the framework to work effectively for this task. Overall comparisons and ablations in the paper, as well as ablations in W2, demonstrate that the proposed face-specific design leads to more structured facial geometry and more consistent textures.
>
> **W2**: The pretrained models serve as foundation components in our pipeline, but the improvement does not come from their strength  alone. We further add ablations to clarify the source of the gains: (1) Generating meshes with an unfinetuned Stage-1 generator equipped with DINOv2 but lacking our face-domain adaptation, with results shown in the following table. (2) Generating textures directly using the foundation geometry-conditioned texturing diffusion model without our facial multi-view prior injection or mesh-image alignment, with results reported in the table in our response to Reviewer Les7, W5. These ablations show that the pretrained components alone do not deliver the desired performance, and that the main gain comes from the face-specific design in SAMT. We will include these ablations in the revision to make the contribution more explicit.
>
>  | Models|ULIP-I↑|ULIP-T↑|Uni3D-I↑|Uni3D-T↑|CD↓|
> |-------------------------|-------:|-------:|--------:|--------:|------:|
> | w/o Adaptation|0.133|0.065|0.291|0.229|0.120|
> | SAMT|**0.149**|**0.084**|**0.323**|**0.251**|**0.105**|
>
> **W3**: The full SAMT pipeline requires ~20 GB GPU memory for inference and takes 129.7 s/sample on a single RTX 4090 GPU, including 21.0 s for mesh generation, 10.8 s for mesh-image alignment (7.5s for facial multi-view prior generation and 3.3s for optimization), and 97.9 s for texture synthesis. The face-domain training requires 8× A100 GPUs (40 GB) for approximately 40 hours. Compared with Hunyuan3D 2.1 at 124.1 s/sample and Trellis 2 at 508.6 s/sample, SAMT achieves comparable inference efficiency. We will add these efficiency and hardware details in the revision.
>
> **W4**: (1) Thanks for pointing this out and we apologize for the misleading claim. In our pipeline, the final mesh is extracted from the decoded occupancy field using Marching Cubes, and no retopology step is performed. Therefore, we do not intend to claim artist-ready or quad-based production topology. Our original intent was to convey that, compared with existing single-image 3D generators, SAMT is able to produce facial meshes with more complete and stable geometry and more coherent structure, making them more suitable for re-rendering and downstream texturing. We will tone down this terminology and replace these terms with more precise descriptions. (2) A current technical limitation is that extreme lighting may be baked into the texture, and heavy occlusions may degrade texture consistency and geometric fidelity. Failure cases are shown in Figure 1 in the anonymous link https://anonymous.4open.science/r/anonymous-24E6, suggesting that this issue is not unique to SAMT, but a common challenge for single-image 3D facial asset generation more broadly. Even in these cases, SAMT still produces relatively more stable facial geometry than existing approaches, benefiting from our face-domain training on the curated facial dataset. We will discuss limitations and add the shown failure cases in the revision.
>
> **Q1**: Please refer to W3.
>
> **Q2**: SAMT is generally robust to moderate real-world inputs, but may become less robust on faces with accessories such as glasses or hats (The last three rows in Figure 2 in the anonymous link). We believe this is mainly because our curated face data is dominated by unobstructed facial examples, while cases involving accessories such as glasses or hats are less well covered. We will clarify this limitation more explicitly in the revision.
>
> **Q3**: Please refer to W4(1). Thanks for the suggestion and we will tone down the claims in the revision.
>
> **Limitations**: Thanks for your suggestion. In the revision, we will add a paragraph and visual examples about extreme cases as described in W4(2).

---

> > ### Author Rebuttal · Reviewer_5j24 · 2026-04-02
> >
> > Thanks to the authors for the detailed rebuttal. It resolves my main concerns.
> > However, after reading other reviews i agree that the evaluation is not entirely fair. Authors should add a more detailed discussion why general-purpose 3D models (such as Trellis or Hunyuan) were used instead of Avatar specific approaches including reasons why it wasn't possible, rather than only comparing a model designed specifically for facial recognition with them.
> > Overall, the pipeline engineering is solid and yields good results. I am raising my score to 4 (weak accept).

---

> > > ### Author Response · Authors · 2026-04-02
> > >
> > > We sincerely appreciate your constructive feedback and are glad that our clarifications helped address the main concerns. We will clarify the evaluation setting more explicitly in the revision as suggested. Our current baselines were chosen because they are recent methods with a comparable explicit 3D asset generation objective, which makes them suitable for evaluating the effect of our face-specific specialization. At the same time, avatar-specific approaches deserve clearer discussion. Many such methods differ from SAMT in representation, inference procedure, and optimization/evaluation target. Some rely on 3DGS or neural representations rather than explicit textured meshes, some use per-instance optimization rather than feed-forward generation pipelines, and some are primarily optimized and evaluated for avatar rendering quality, reenactment, or animatability rather than explicit mesh-texture asset quality. For this reason, these methods do not always yield a comparable output form and are therefore not always directly comparable to SAMT, whose goal is to generate an explicit facial mesh with multi-view-consistent texture from a single image.
> > >
> > > Your suggestions are highly valuable for improving the quality of our work. We will incorporate all the additional discussions and experiments in the rebuttal into the relevant parts of the revised final manuscript. We thank you again for your time and helpful suggestions.

---

### Official Review · Reviewer_Les7 · 2026-03-16

**Soundness:** 3
**Presentation:** 3
**Significance:** 2
**Originality:** 2
**Overall Recommendation:** 5
**Confidence:** 3

**Summary:**

The paper proposes SAMT, a two-stage pipeline for generating production-ready 3D facial assets from a single image. Stage 1 adapts an image-conditioned 3D latent diffusion model (DiT with a DoraVAE latent) via large-scale, face-domain fine-tuning (35k curated meshes) to produce topology-regular, detailed facial meshes. Stage 2 synthesizes multiview-consistent textures by injecting facial multi-view priors from a specialized 2D generator into a geometry-conditioned 2D texturing diffusion, coupled with a derivative-free mesh–image alignment step. Experiments against recent 3D generators and texturing methods indicate improved facial geometry structure and finer, more view-consistent textures.

**Compliance With Llm Reviewing Policy:**

Affirmed.

**Key Questions For Authors:**

1. What is the exact architecture and training data of the “pretrained geometry-conditioned 2D texturing diffusion” used in Stage 2? Can you provide training details, conditioning pathways, and multi-view attention implementation to ensure reproducibility?
2. Why is mesh–image alignment restricted to scale and translation? Did you try optimizing rotation, or at least yaw-only rotation, and how did it affect consistency and identity?
How is UV unwrapping handled for the final textured asset? Is the UV layout standardized across outputs, and do you ensure minimal stretching/seams? Do you generate only albedo, or any PBR channels (normal/roughness/specular)?
3. How robust is the method to extreme lighting and occlusions in the input image? Do you bake illumination into the texture or attempt any intrinsic decomposition?
4. For the geometry evaluation with ULIP/Uni3D and GPT-5.2-generated texts, can you detail prompts, seeds, and release the description set for reproducibility? How sensitive are the conclusions to the choice of LLM or textual prompts?

**Limitations:**

yes

**Strengths And Weaknesses:**

Strengths
1. Leveraging a face-specialized fine-tuning regime (joint DiT + LoRA) on a sizable curated head dataset for robust, structured facial mesh generation.
2. A practical multiview-aware texturing strategy that explicitly injects multi-view facial priors into a geometry-conditioned diffusion pipeline, alongside a derivative-free (Powell) alignment step to improve identity fidelity and cross-view consistency.
3. Sensible adoption of DoraVAE with sharp-edge sampling and dual cross-attention to better preserve facial micro-structures in latent space.
4. Clear high-level pipeline description, stage-wise figures, and straightforward articulation of the mesh–image alignment objective.
5. Addresses a highly impactful problem: single-image to high-quality, re-renderable 3D face assets with improved structural regularity and texture consistency.
6. The approach offers a pragmatic path that many in the community can build upon for production applications in AR/VR and content creation.

Weaknesses
1. Limited methodological novelty in Stage 1: largely a careful specialization of existing latent 3D diffusion backbones (DiT + DoraVAE) and LoRA fine-tuning rather than a fundamentally new 3D generation method.
2. The mesh–image alignment optimizes only global scale and translation; omitting rotation could be fragile if canonical orientation is imperfectly aligned to the multi-view priors.
3. Geometry evaluation mix includes non-standard cross-modal proxy metrics (ULIP/Uni3D) and a small-scale CD evaluation; more robust geometry comparisons with paired ground truth would strengthen claims.
4. Use of GPT-5.2 to produce textual descriptions for retrieval metrics undermines reproducibility and may bias results; details on prompt design and reproducibility are missing.
5. The pretrained “geometry-conditioned 2D texturing diffusion” is not specified (architecture, training data, resolution), limiting reproducibility and clarity on what capacity is contributed by SAMT vs the foundation texturer.
6. Recent face/UV texturing pipelines like AvatarTex (diffusion-to-GAN over UVs, multi-stage alignment) are not discussed, despite conceptual overlap in identity preservation and cross-view coherence.

---

> ### Author Rebuttal · Authors · 2026-03-30
>
> We sincerely thank the reviewer for the thoughtful and constructive feedback, and we respond to the comments point by point below.
>
> **W1**: We would like to clarify that Stage 1 is **not** intended as a fundamentally new general-purpose 3D generation backbone. Instead, it targets an under-explored task that remains difficult for existing methods: explicit 3D facial asset generation from a single image, where prior methods often produce inaccurate facial geometry and irregular mesh structures. In contrast, SAMT achieves more stable and faithful facial geometry generation through face-specialized adaptation of the latent 3D diffusion on our curated 35K facial dataset, as demonstrated by the experimental results in Table 1 and Figure 5. Therefore, we believe that the novelty of Stage 1 lies in providing an effective solution for generating structured and detailed facial meshes in this challenging single-image setting.
>
> **W2**: After the face-domain finetuning, our 3D diffusion model generates meshes in a stable frontal view, and the facial priors are synthesized under the same fixed canonical orientation. Thus, the dominant mismatch is scale and translation rather than rotation. We also tested a yaw-aware alignment optimization (s+t+yaw), but it did not yield noticeable gains, as shown in Table 1 in the anonymous link https://anonymous.4open.science/r/anonymous-24E6. Therefore did not optimize rotation.
>
> **W3**: Thanks for this suggestion. Table 2 in the anonymous link provides additional F1-score (threshold 0.01) evaluation with ground truth, which will be included in Table 1 in the revised paper. Results also support our conclusion that SAMT provides more faithful facial geometry than existing approaches.
>
> **W4**: We will provide the mentioned generation settings in the revision and release our code for reproducibility. We also conduct an additional sensitivity analysis using different LLMs and prompts in Table 3 in the anonymous link. Each entry reports the ULIP-T↑ / Uni3D-T↑, and P1 and P2 denote two different prompt templates. Results show that the relative ranking and overall conclusions are not materially affected by the choice of LLM or prompt wording, indicating that our conclusions are not sensitive to them. We will include these results in the revised version.
>
> **W5**: The texturer uses a Stable-Diffusion-style architecture, trained on filtered textured assets in Objaverse-XL at 512×512 resolution. It concatenates geometric conditions with latent noise and injects reference-image features into the denoising network. Its multi-view attention is implemented through cross-view latent feature aggregation. We add ablations that compare the base texturer with SAMT in the following Table. Results indicate that the final gains come from our face-specialized design rather than the application of the foundation texturer.
>  |     | CLIP-score↑ | FID-CLIP↓ | CMMD↓ | ArcFace↓ |
> |---------------------------|-----------:|---------:|------:|--------:|
> | Foundation texturing diffusion  | 0.823      | 33.410   | 2.985 | 0.329  |
> | w. facial multi-view prior| 0.833      | 32.406   | 2.821 | 0.319   |
> | w. mesh-image alignment   | 0.828      | 33.173   | 2.980 | 0.324   |
> | SAMT | **0.839**  | **32.211** | **2.805** | **0.316** |
>
> **W6**: We will expand the related work to include recent face/UV texturing pipelines such as AvatarTex. AvatarTex is a face-specific UV-space reconstruction pipeline tailored to standardized UV-space facial texture reconstruction, while SAMT textures generated meshes without a fixed UV layout and addresses the limitation of UV-space methods (described in Line 50-54) through our proposed facial-multi-view aware texturing strategy. We will revise the related work to make the distinctions clearer.
>
> **Q1**: Please refer to W5.
>
> **Q2**: Please refer to the reply in W2 regarding rotation. UV unwrapping is performed automatically using xatlas. The UV layout is not standardized but mesh-specific, and we do not explicitly optimize UV stretching or seam visibility. The texturer predicts albedo and metallic/roughness maps. We apologize for the confusion and will add these details in the revision.
>
> **Q3**: (1) SAMT is reasonably robust to moderate lighting and occlusions, but extreme lighting and severe occlusions remain challenging. This is not unique to SAMT, but a common difficulty in single-image 3D facial asset generation. As shown in the top three rows of Figure 1 in the anonymous link, extreme lighting may affect texture consistency, and severe occlusions may further reduce geometric fidelity. Despite this, SAMT is relatively more robust on geometry compared to existing methods, which we attribute to the face-domain specialization on our curated facial data in Stage 1. (2) Sorry for the confusion. We do not explicitly perform intrinsic decomposition, so illumination may still be baked into the texture. We will clarify this limitation in the revision.
>
> **Q4**: Please refer to W4.

---

> > ### Author Rebuttal · Reviewer_Les7 · 2026-04-03
> >
> > Thank you to the authors for the detailed rebuttal and the additional ablations provided. The proposed pipeline is highly practical for the community. I will maintain my score and strongly recommend incorporating these comprehensive ablations and limitation discussions into the final manuscript.

---

> > > ### Author Response · Authors · 2026-04-06
> > >
> > > We sincerely thank you for the encouraging feedback and constructive suggestions, which are of great help in refining our manuscript. Following your suggestions, we will incorporate the additional experiments and discussions into the relevant parts of the revised manuscript to further strengthen the paper.

---

### Decision · Program_Chairs · 2026-04-30

**Decision:**

Accept (regular)

**Comment:**

This paper initially received mixed reviews (5, 3, 3). After the rebuttal, most of the concerns were sufficiently addressed, and both negative reviewers increased their scores from 3 to 4, resulting in unanimously positive final scores.

The reviewers agree that the paper addresses an important problem and the proposed solution is well-motivated and effective. Concerns were raised in the reviews regarding the novelty of the proposed method and the evaluation and comparison, in particular, the lack of comparison against avatar-specific reconstruction methods. The rebuttal addresses theses concerns with clarifications and new comparison results. Overall, the Area Chairs agree with the reviewers that the strengths outweigh the weaknesses and therefore recommend accepting the manuscript.